# Altered firing output of VIP interneurons and early dysfunctions in CA1 hippocampal circuits in the 3xTg mouse model of Alzheimer's disease

**Felix Michaud[1,2†], Ruggiero Francavilla[1,2†‡, §], Dimitry Topolnik[1,2], Parisa Iloun[1,2], Suhel Tamboli[1,2], Frederic Calon[2,3], Lisa Topolnik[1,2]***

[1]Department of Biochemistry, Microbiology and Bio-informatics, Laval University, Québec, Canada; [2]Neuroscience Axis, CHU de Québec Research Center (CHUL), Québec, Canada; [3]Faculty of Pharmacy, Laval University, Quebec, Canada

**\*For correspondence:**
lisa.topolnik@bcm.ulaval.ca

[†]These authors contributed equally to this work

**Present address:** [‡]CHU Sainte-Justine Research Center, Université de Montréal, Montreal, Canada; [§]Department of Neurosciences, Université de Montréal, Montreal, Canada

**Competing interest:** The authors declare that no competing interests exist.

**Abstract** Alzheimer's disease (AD) leads to progressive memory decline, and alterations in hippocampal function are among the earliest pathological features observed in human and animal studies. GABAergic interneurons (INs) within the hippocampus coordinate network activity, among which type 3 interneuron-specific (I-S3) cells expressing vasoactive intestinal polypeptide and calretinin play a crucial role. These cells provide primarily disinhibition to principal excitatory cells (PCs) in the hippocampal CA1 region, regulating incoming inputs and memory formation. However, it remains unclear whether AD pathology induces changes in the activity of I-S3 cells, impacting the hippocampal network motifs. Here, using young adult 3xTg-AD mice, we found that while the density and morphology of I-S3 cells remain unaffected, there were significant changes in their firing output. Specifically, I-S3 cells displayed elongated action potentials and decreased firing rates, which was associated with a reduced inhibition of CA1 INs and their higher recruitment during spatial decision-making and object exploration tasks. Furthermore, the activation of CA1 PCs was also impacted, signifying early disruptions in CA1 network functionality. These findings suggest that altered firing patterns of I-S3 cells might initiate early-stage dysfunction in hippocampal CA1 circuits, potentially influencing the progression of AD pathology.

## eLife assessment

This study describes **fundamental** findings related to early disruptions in disinhibitory modulation exerted by VIP+ interneurons, in CA1 in a transgenic model of Alzheimer's disease pathology. The authors provide a **compelling** analysis at the cellular, synaptic, network, and behavioral levels on how these changes correlate and might be related to behavioral impairments during these early stages of AD pathology.

## Introduction

Memory impairments pose a significant burden on individuals grappling with Alzheimer's disease (AD). A series of recent investigations has underscored the pivotal role of hippocampal GABAergic interneurons (INs) in memory dysfunction in AD (*Verret et al., 2012*; *Cattaud et al., 2018*; *Martinez-Losa et al., 2018*; *Schmid et al., 2016*; *Rice et al., 2020*; *Caccavano et al., 2020*; *Chung et al., 2020*; *Hijazi et al., 2020*; *Algamal et al., 2022*; *Hernández-Frausto et al., 2023*). Among the various types of hippocampal INs, CA1 parvalbumin- (PV-INs), somatostatin- (SOM-INs), and to a certain extend

calretinin- (CR-INs) expressing cells have received significant attention. Postmortem data reveal diminished numbers of SOM-INs in the temporal cortex of AD patients, while the populations of PV-INs and CR-INs remain relatively unaffected (*Hof et al., 1993*; *Brady and Mufson, 1997*; *Waller et al., 2020*). Furthermore, significant synaptic rewiring in SOM-INs due to a decreased cholinergic drive has been documented in mice with AD-like pathology (*Schmid et al., 2016*). This disturbance has led to memory deficits, underscoring the central role played by SOM-INs in the dysfunction observed in AD (*Schmid et al., 2016*). Whereas most INs control different stages of hippocampal information processing by providing inhibition to different subcellular domains of principal cells (PCs), a specific population of the vasoactive intestinal polypeptide-expressing cells (VIP-INs) regulates the activity of INs (*Acsády et al., 1996*; *Tyan et al., 2014*; *Francavilla et al., 2015*; *Francavilla et al., 2018*; *Luo et al., 2019*; *Luo et al., 2020*; *Kullander and Topolnik, 2021*). The hippocampal CA1 VIP-INs engage in goal-directed and spatial learning by tuning the activity of PCs via input-specific disinhibition (*Magnin et al., 2019*; *Turi et al., 2019*; *Bilash et al., 2023*). Specifically, long-range glutamatergic inputs from the lateral entorhinal cortex can recruit VIP-INs to modulate dendritic excitability of CA1 PCs through disinhibition and support contextual memory (*Bilash et al., 2023*). The role of circuit disinhibition and VIP-INs in the AD pathology remains unknown.

The 3xTg-AD (Tg(APPSwe,tauP301L)1Lfa *Psen1^{tm1Mpm}*/Mmjax) mouse model, combining three familial AD mutations (Swedish *APP* KM670/671NL, *PSEN1* M146V, and *MAPT P301L*) exhibits accumulation of amyloid-β (Aβ) plaques and neurofibrillary tangles, synaptic impairment, and cognitive deficits (*Javonillo et al., 2021*). As a result, 3xTg-AD mutant mice have become a primary choice for investigating AD mechanisms (*Drummond and Wisniewski, 2017*; *Myers and McGonigle, 2019*). Extensive work employing this model has focused on the CA1 region of the hippocampus, recognized as a central hub for spatial learning and episodic memory (*Oddo et al., 2003*; *Chakroborty et al., 2009*; *Noristani et al., 2011*; *Clark et al., 2015*; *Mably et al., 2017*; *Stimmell et al., 2019*; *Javonillo et al., 2021*). In vitro and in vivo studies with 3xTg-AD mutants have provided evidence of disrupted synaptic plasticity in CA1 as one of the earliest signs of pathology. This occurs even before the accumulation of extracellular Aβ plaques and neurofibrillary tangles, which begins at 6 months of age and progresses further after 12 months (*Davis et al., 2014*; *Clark et al., 2015*; *Javonillo et al., 2021*). While CA1 INs are known to play a crucial role in regulating synaptic plasticity, memory encoding, and consolidation (*Topolnik and Tamboli, 2022*), it is yet unclear whether they undergo early changes in the AD progression. However, there is evidence indicating that mice with overexpression of the amyloid precursor protein (APP) with the Swedish mutation and deltaE9 mutation in presenilin 1 (*APP/PS1*) display reduced activity of CA1 PV-INs but increased activity of SOM-INs (*Algamal et al., 2022*). This suggests that disrupted communication within local inhibitory circuits may contribute to APP/PS1-mediated memory dysfunction (*Palop and Mucke, 2016*). Nevertheless, investigations into the activity profiles and local synaptic communication among CA1 INs of 3xTg-AD mice in vivo have been lacking. Importantly, prior studies on aged mice have identified significant changes in firing properties of type 3 interneuron-specific interneurons (I-S3), which co-express CR and VIP and control the activity of SOM-INs but also of PV-INs in the CA1 hippocampal region (*Tyan et al., 2014*; *Francavilla et al., 2020*; *Luo et al., 2020*). Whether impairments in I-S3 activity might contribute to altered CA1 inhibition and network function in 3xTg-AD mice remains unknown.

In this study, we examined IN function in the CA1 region of young adult (P90–260) 3xTg-AD mice. In awake animals, CA1 INs displayed increased activity, specifically during tasks involving spatial decision-making and object exploration. This increased activity of INs was not a result of changes in the excitatory drive to INs but rather stemmed from a reduced inhibitory tone, indicating impaired inhibition of CA1 INs. Consequently, the firing rate of I-S3 cells, which provide inhibition to CA1 INs, was significantly reduced. At a network level, these alterations in IN function had an impact on the activation of CA1 PCs, the fundamental units for processing information in the hippocampus (*Soltesz and Losonczy, 2018*). Thus, these early changes may contribute to an imbalance between excitation and inhibition, impairing induction of synaptic plasticity, and ultimately leading to memory dysfunction.

## Results

### VIP-INs accumulate intracellular Aβ and its precursor at the early stages of AD progression

To examine the properties of VIP/CR-coexpressing I-S3 cells in AD-afflicted mice, we generated VIP-eGFP-3xTg (VIP-Tg) mice by crossing homozygous 3xTg-AD (Tg) with VIP-eGFP (VIP-nonTg) mice, in which all INs immunoreactive for VIP express eGFP (*Tyan et al., 2014*). To pinpoint the stage with initial signs of pathology in these experimental animals, we first assessed Aβ expression in sections from the VIP-Tg animals using the 6E10 antibody which binds to an epitope found in both human APP and Aβ and may detect full-length APP, α-APP (soluble alpha-secretase-cleaved APP), the C-terminal (C99) fragment of APP and Aβ (*LaFerla et al., 2007*). Consistent with previous findings (*Mastrangelo and Bowers, 2008*; *Davis et al., 2013*), we observed widespread intracellular Aβ/APP labeling within the CA1 pyramidal layer from P90, progressing further by P180 (*Figure 1a*; VIP-Tg: *n* = 8 sections/3 animals; *Supplementary file 1*). Moreover, intracellular Aβ/APP accumulation was observed in VIP-INs with soma located within *stratum pyramidale* (PYR) and *stratum radiatum* (RAD) (*Figure 1b*).

At this age, Tg mice are typically considered asymptomatic as they do not show major memory impairments (*Pairojana et al., 2021*; *Javonillo et al., 2021*). However, previous reports suggest episodic-like memory deficits in 3- to 6-month-old Tg mice when using complex tests combining object structure, location, and context for assessment of recognition memory (*Davis et al., 2013*; *Davis et al., 2014*). To examine whether this applies to VIP-Tg animals, we compared the performance of VIP-nonTg vs. VIP-Tg animals in the object-place test (OPT; *Figure 1—figure supplement 1a, b*; VIP-nonTg: *n* = 10 mice; VIP-Tg: *n* = 11 mice; *Supplementary file 1*; *Francavilla et al., 2020*), and observed a significant impairment in the recognition index (RI) for novel object. Notably, the total object exploration time and the overall animal activity did not significantly differ between the two mouse lines (*Figure 1—figure supplement 1c*; VIP-nonTg: *n* = 10 mice; VIP-Tg: *n* = 11 mice; *Supplementary file 1*), suggesting that the lower RI in VIP-Tg mice is associated with early deficits in novelty recognition rather than decreased exploratory behavior. These findings indicate that VIP-Tg mice show initial signs of AD-related pathology with intracellular Aβ accumulation in PCs and VIP-INs, along with associated deficits in episodic-like memory starting from P90. Accordingly, for I-S3 analysis, we selected mice aged between P90 and P180 (*n* = 27).

To explore the potential loss of CA1 VIP INs in VIP-Tg mice, we conducted immunohistochemistry on hippocampal slices from both VIP-nonTg and VIP-Tg mice (*Figure 1c, d*; *Supplementary file 1*). We observed no alteration in the density of VIP-INs within the CA1 region (*Figure 1c, d*; VIP-nonTg: *n* = 17 sections/3 animals; VIP-Tg: *n* = 13 sections/3 animals), suggesting that this interneuronal population persists in VIP-Tg mice. Additionally, our findings revealed no disparity in either CR-INs or VIP subtype co-expressing CR and VIP and representing the I-S3 cells (*Figure 1c, d*; VIP-nonTg: *n* = 11 sections/3 animals; VIP-Tg: *n* = 11 sections/3 animals), although a significant loss of I-S3 cells was apparent when removing the outlier (Wilcoxon rank test, p = 0.0193; VIP-nonTg *n* = 11 sections/3 animals; VIP-Tg: *n* = 10 sections/3 animals). This indicates that I-S3 cells still survive in young VIP-Tg mice but their density tends to decrease upon pathology progress.

### I-S3 cells exhibit wider spikes but lower firing rate in VIP-Tg mice

Given that AD pathology can influence the firing behavior of hippocampal neurons (*Vitale et al., 2021*; *Olah et al., 2022*), we proceeded with targeted patch-clamp recordings from I-S3 cells to compare their passive and active membrane properties in slices obtained from VIP-nonTg and VIP-Tg mice (*Figure 2*). Notably, the resting membrane potential, input resistance, membrane capacitance, and membrane time constant of I-S3 cells showed no significant differences between VIP-nonTg (−60.5 ± 1.7 mV, 517 ± 37 MΩ, 32.8 ± 3.5 pF, 36.5 ± 3.2 ms) and VIP-Tg mice (−65.8 ± 2.1 mV, 493 ± 45 MΩ, 29.3 ± 2.9 pF, 35.7 ± 2.2 ms), respectively (VIP-nonTg: *n* = 8 cells/5 animals; VIP-Tg: *n* = 10 cells/3 animals). However, distinct alterations were evident in firing and action potential (AP) properties (*Figure 2*). Specifically, I-S3 cells in VIP-Tg animals exhibited a significant increase in AP half-width and area, along with a slower AP repolarization phase compared to VIP-nonTg mice (*Figure 2a–c*; VIP-nonTg: *n* = 8 cells/5 animals; VIP-Tg: *n* = 10 cells/3 animals; *Supplementary file 1*). Moreover, at a 2× rheobase current during the firing train, although the AP amplitude was lower in VIP-Tg mice, the prolonged AP half-width resulted in an increased AP area (*Figure 2d*; VIP-nonTg: *n* = 7 cells/5 animals; VIP-Tg: *n* = 7 cells/3 animals; *Supplementary file 1*). The elongated AP duration linked to a

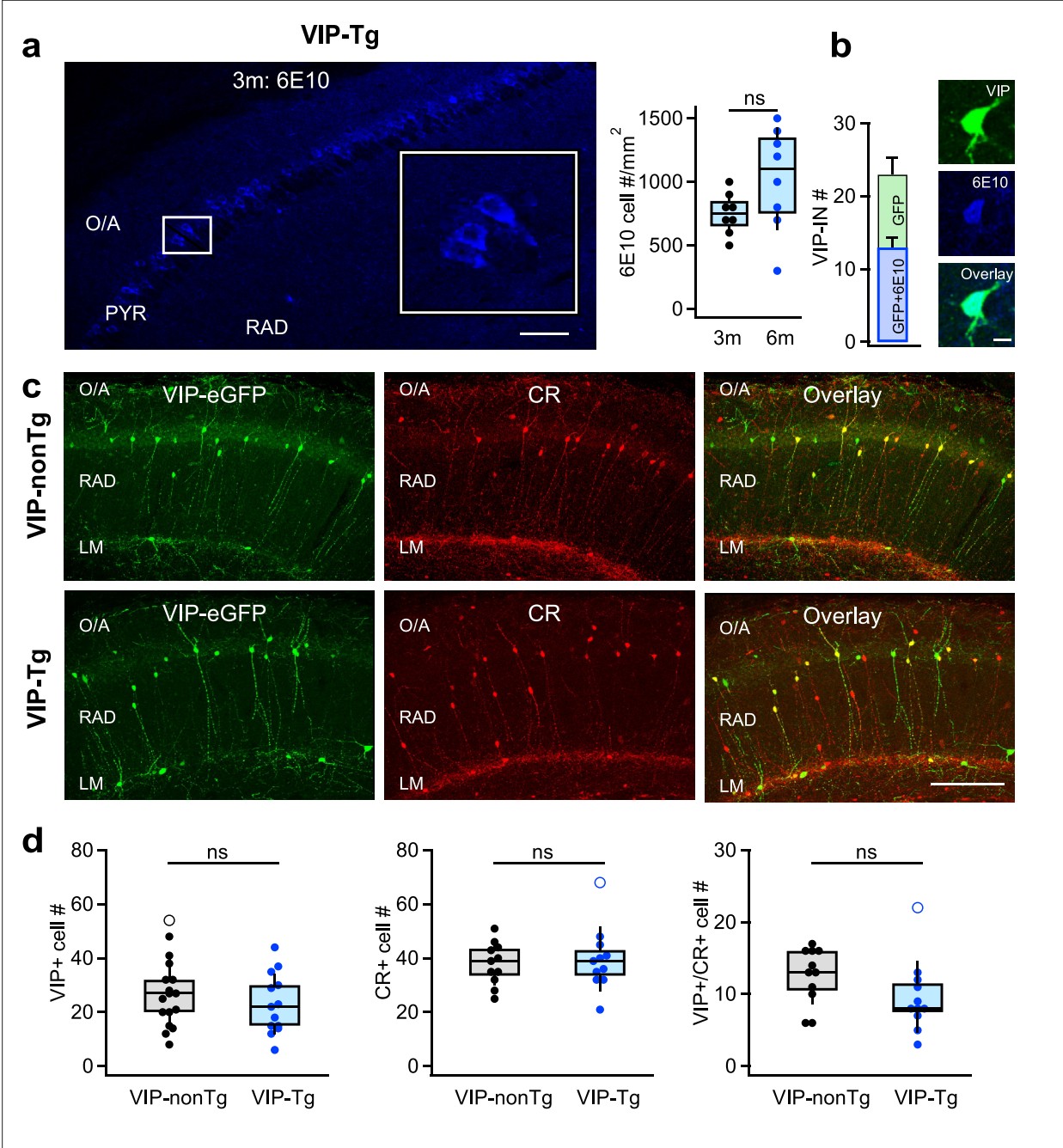

**Figure 1.** VIP/CR I-S3 cells survive but accumulate intracellular amyloid-β (Aβ) and its precursors in VIP-Tg mice. (**a**) Confocal image of the CA1 hippocampal area showing intracellular accumulation of amyloid precursor protein (APP)/Aβ (6E10) in cell bodies of the pyramidal layer in a 3-month-old VIP-Tg mouse (left, scale bar: 100 μm; inset shows the area indicated with a white box) and summary box plot showing the density of cells expressing APP/Aβ intracellularly in 3- vs. 6-month-old mice (right; $n$ = 8 slices from 3 animals per group). (**b**) Summary bar graphs showing the average number of VIP+ cells expressing APP/Aβ intracellularly per slice (left), and representative confocal images of the VIP-IN labeled with eGFP and showing intracellular accumulation of APP/Aβ in a 6-month-old VIP-Tg mouse (right). Scale bar: 10 μm (**c**) Representative confocal images of the CA1 hippocampal area in VIP-nonTg (top) and VIP-Tg (bottom) mice showing eGFP expression in VIP INs (left), CR expression (middle), and an overlay of both markers (right). Scale bar: 100 μm. (**d**) Summary plots showing no changes in the number of VIP-INs (left; VIP-nonTg: $n$ = 17 slices/3 animals; VIP-Tg: $n$ = 13 slices/3 animals), CR-INs (middle; VIP-nonTg: $n$ = 11 slices/3 animals; VIP-Tg: $n$ = 11 slices/3 animals), and VIP/CR co-expressing I-S3 cells (right; VIP-nonTg: $n$ = 11 slices/3 animals, VIP-Tg: $n$ = 11 slices/3 animals) in VIP-Tg mice.

The online version of this article includes the following figure supplement(s) for figure 1:

**Figure supplement 1.** Early deficits in the object-place test (OPT) of VIP-Tg mice.

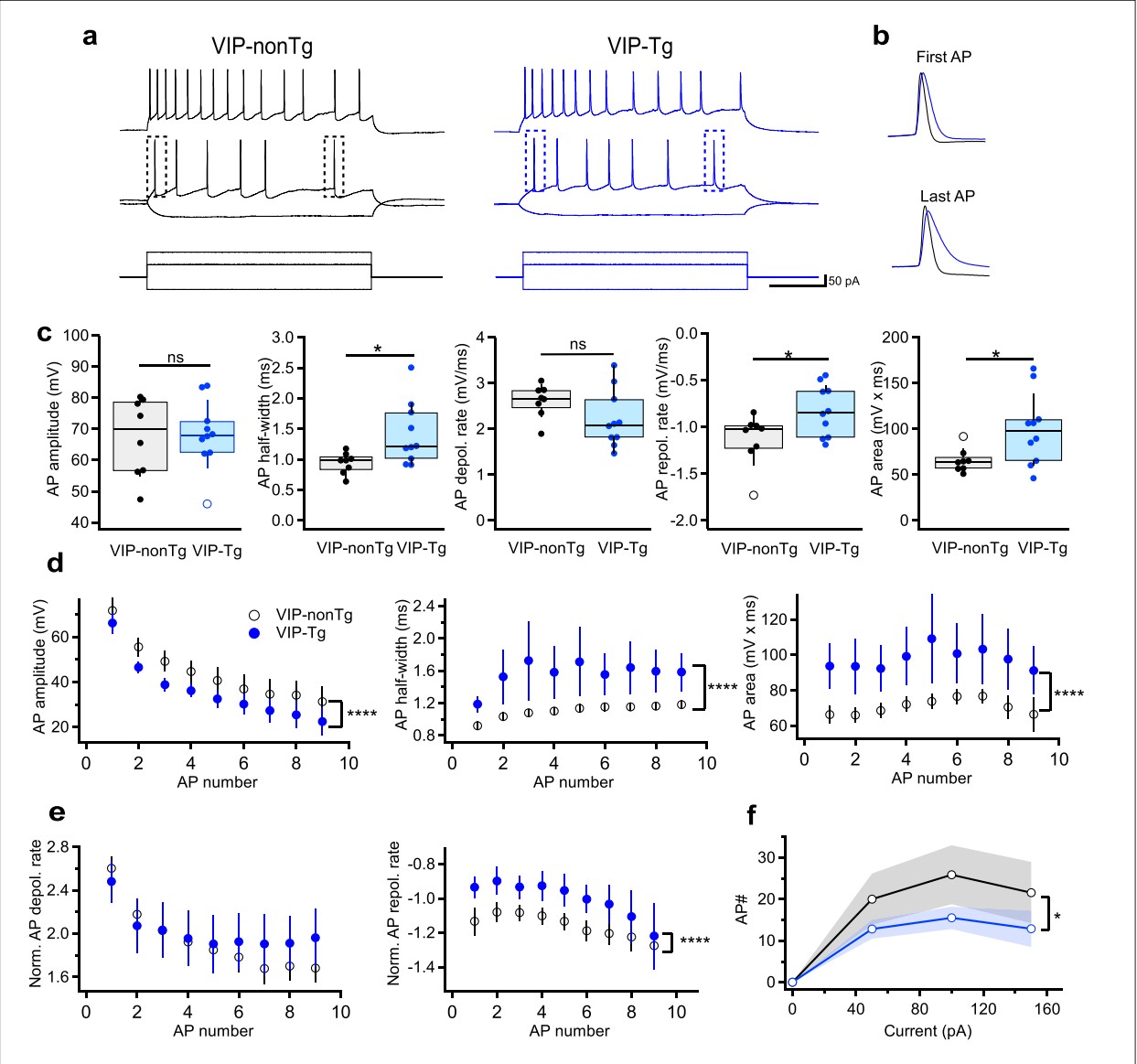

**Figure 2.** Cells exhibit wider spikes and lower firing rate in VIP-Tg mice. (**a**) Representative voltage responses (top, middle) showing the firing pattern of I-S3 cells in VIP-nonTg (left, black) and VIP-Tg (right, blue) mice in response to depolarizing and hyperpolarizing current injections (bottom). (**b**) Representative voltage traces in VIP-nonTg (black) and VIP-Tg (blue) mice superimposed for the first (top) and last (bottom) AP showing the longer AP duration in VIP-Tg mice. (**c**) Summary plots showing the amplitude (left), half-width (middle left), depolarization rate (middle), repolarization rate (middle right), and area (right) of the first AP in I-S3 cells. *p < 0.05 (VIP-nonTg: $n$ = 8 cells/5 mice; VIP-Tg: $n$ = 10 cells/3 mice). (**d**) Summary plots showing changes in the AP amplitude (left), half-width (middle), and area (right) during the train of APs in I-S3 cells. ****p < 0.0001 (VIP-nonTg: $n$ = 7 cells/5 mice; VIP-Tg: $n$ = 7 cells/3 mice). (**e**) Summary plots showing no changes in the AP depolarization rate (left) but slower repolarization rate (right) during the train of APs in I-S3 cells. ****p < 0.0001 (VIP-nonTg: $n$ = 7 cells/5 mice; VIP-Tg: $n$ = 7 cells/3 mice). (**f**) Summary plot showing changes in the number of APs during the train of APs in relation to the injected current in I-S3 cells. The shaded areas show the standard error of the mean (SEM). *p < 0.05 (VIP-nonTg: $n$ = 7 cells/5 mice; VIP-Tg: $n$ = 7 cells/3 mice).

The online version of this article includes the following figure supplement(s) for figure 2:

**Figure supplement 1.** Unaltered synaptic drive to I-S3 cells in VIP-Tg mice.

**Figure supplement 2.** Unaltered morphological properties of I-S3 cells in VIP-Tg mice.

slower AP repolarization rate led to a reduced firing rate of I-S3 cells in VIP-Tg animals (***Figure 2e, f***; VIP-nonTg: $n$ = 7 cells/5 animals; VIP-Tg: $n$ = 7 cells/3 animals; ***Supplementary file 1***). Collectively, these findings highlight specific alterations in the firing output of I-S3 INs, characterized by prolonged individual spikes and reduced firing frequency in the early stages of AD pathology.

To investigate whether alterations in physiological properties extend to changes in synaptic input that I-S3 cells receive, we conducted recordings of spontaneous inhibitory (sIPSCs) and excitatory (sEPSCs) postsynaptic currents in slices from VIP-nonTg and VIP-Tg mice (*Figure 2—figure supplement 1*). However, our findings revealed no significant difference in either the amplitude or frequency of sIPSCs (*Figure 2—figure supplement 1a–c*;s VIP-nonTg: $n$ = 7 cells/5 animals; VIP-Tg: $n$ = 6 cells/5 animals; *Supplementary file 1*). Likewise, both the amplitude and frequency of sEPSCs displayed no noticeable changes (*Figure 2—figure supplement 1a–c*; VIP-nonTg: $n$ = 5 cells/5 animals; VIP-Tg: $n$ = 4 cells/4 animals; *Supplementary file 1*). Consequently, these results suggest that the synaptic inputs to I-S3 cells remain unaffected in VIP-Tg mice.

Given the potential for hippocampal CR-INs to exhibit abnormal morphology characterized by prominent dystrophic neurites due to Aβ accumulation (*Baglietto-Vargas et al., 2010*), we also investigated the morphological properties of anatomically confirmed I-S3 INs reconstructed in Neurolucida (*Figure 2—figure supplement 2*). However, our analysis revealed no variations in the morphology of I-S3 cells. Specifically, there were no differences observed in soma area, dendritic surface, total dendritic length, or the number of branching points (*Figure 2—figure supplement 2b–e*; VIP-nonTg:

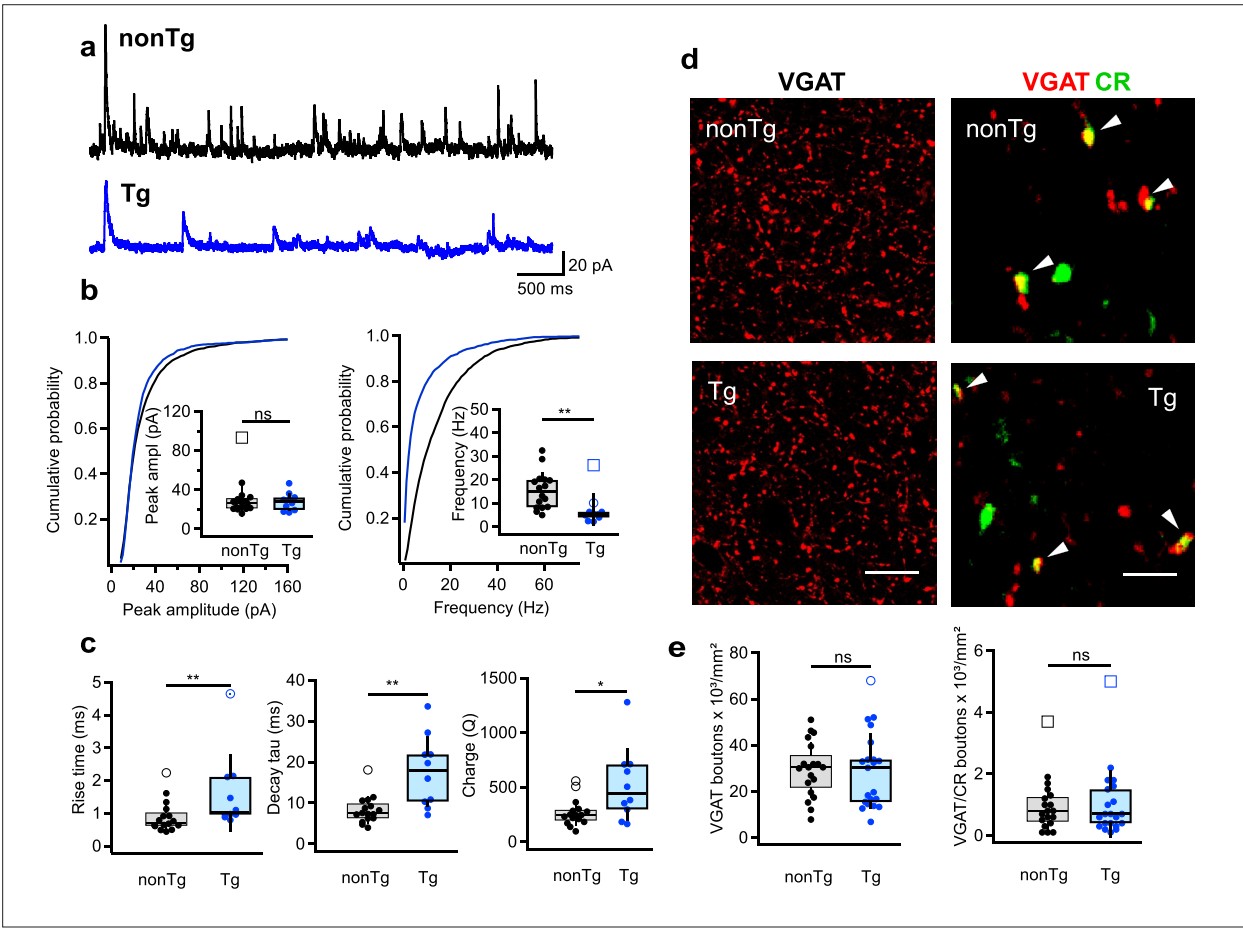

**Figure 3.** Altered inhibition of CA1 interneurons in Tg mice. (**a**) Representative traces for spontaneous inhibitory postsynaptic currents (sIPSCs) in *stratum oriens/alveus* (O/A) INs from nonTg (top, black) and Tg (bottom, blue) mice. (**b**) Summary plots showing the sIPSCs amplitude (left) and frequency (right) in nonTg (black) and Tg (blue) groups. Insets show data for animal-wise comparison. **p < 0.01 (nonTg: $n$ = 16 cells/5 mice; Tg: $n$ = 10 cells/4 mice). (**c**) Summary plots showing the sIPSC rise time left, decay time constant (middle), and charge transfer (right) in O/A INs of nonTg and Tg animals. *p < 0.05, **p < 0.01 (nonTg: $n$ = 16 cells/5 mice; Tg: $n$ = 10 cells/4 mice). (**d**) Representative confocal images showing immunoreactivity for vesicular GABA transporter (VGAT) (left, scale bar: 20 µm) and VGAT + CR (right, scale bar: 5 µm) in the CA1 O/A of the nonTg (top) and Tg (bottom) mice. (**e**) Summary plots showing no changes in the density of VGAT- and VGAT/CR-expressing axonal boutons in the O/A (nonTg: $n$ = 19 slices/4 animals; Tg: $n$ = 21 slices/4 animals).

The online version of this article includes the following figure supplement(s) for figure 3:

**Figure supplement 1.** Excitatory drive to *stratum oriens/alveus* (O/A) INs remains unchanged in Tg mice.

*n* = 6 cells/5 mice; VIP-Tg: *n* = 5 cells/4 mice; *Supplementary file 1*). Moreover, the dendritic Sholl analysis displayed no disparities in the number of dendritic intersections, dendritic length, or the number of dendritic nodes at various distances from the soma (*Figure 2—figure supplement 2f*; VIP-nonTg: *n* = 5 cells/5 mice; VIP-Tg: *n* = 5 cells/4 mice). These findings suggest that I-S3 cells only modify their firing properties in VIP-Tg mice without significant alteration in other cellular parameters.

## Altered inhibitory drive to *stratum oriens/alveus* INs in Tg mice: impact of I-S3 cell firing patterns

To examine whether the modified firing properties of I-S3 cells could impact the inhibitory drive to their postsynaptic targets – distinct types of *stratum oriens/alveus* (O/A) INs (*Tyan et al., 2014*), we conducted recordings of sIPSCs in O/A INs (*Figure 3a–c*; nonTg: *n* = 16 cells/5 animals; Tg: *n* = 10 cells/4 animals). Among recorded cells, both OLM and bistratified cells, the main targets of I-S3 cells, were identified using biocytin staining. These data revealed no significant change in the amplitude of sIPSCs, but a reduction in the frequency of sIPSCs events in the Tg group (*Figure 3b*; *Supplementary file 1*). Moreover, our data indicated increased rise and decay time constant of sIPSCs, consequently enhancing the charge transfer (*Figure 3c*; *Supplementary file 1*) in the Tg group.

To explore whether the decreased frequency of inhibitory events in O/A INs might be linked to a loss of synaptic contacts from I-S3 cells, we examined the density of inhibitory axonal boutons labeled with vesicular GABA transporter (VGAT) and CR within CA1 O/A of both groups (*Figure 3d, e*; nonTg: *n* = 19 sections/4 animals; Tg: *n* = 21 sections/4 animals). However, the data showed no alteration in either VGAT or VGAT-CR boutons between the two groups (*Figure 3e*; *Supplementary file 1*). We also examined potential changes in the excitatory drive to O/A INs by recording sEPSCs but found no disparities in the amplitude, frequency, or kinetics of those currents between Tg and nonTg animals (*Figure 3—figure supplement 1a–c*; nonTg: *n* = 8 cells/5 animals; Tg: *n* = 5 cells/4 animals; *Supplementary file 1*). Collectively, these data indicate, that CA1 INs in Tg mice exhibit a modified inhibitory input characterized by slower kinetics and reduced frequency of inhibitory currents. Moreover, these alterations were not a result of modified synaptic connectivity but rather seemed originated from specific changes in the firing output of I-S3 cells, characterized by slower APs and a decreased firing rate.

## Enhanced IN activity during cognitive tasks in Tg mice

To explore whether alterations in the inhibitory drive received by CA1 INs could influence their engagement during behavior, we monitored interneuronal activity during spatial navigation and object encoding. We performed wireless fiber photometry within the CA1 region of the hippocampus in homozygous Tg-mutant mice (*n* = 13) and their nonTg littermates (*n* = 13) while they were awake in their home cages, the T-maze or the open field maze (OFM) (*Figure 4—figure supplement 1*). The Dlx-driven expression of GCaMP6f allowed us to assess the activity of both PV-IN and SOM-IN types – the main targets of I-S3 INs. Given the fiber properties (core: 400 μm; NA: 0.39), the signal was collected from all INs situated within O/A and PYR of the CA1 region (*Pisano et al., 2019*; *Montinaro et al., 2021*).

We observed no changes in the overall behavior of the animals or the activity of INs in either the home cage (nonTg: *n* = 9 animals; Tg: *n* = 8 animals) or OFM (nonTg: *n* = 13 animals; Tg: *n* = 13 animals) in Tg-mutant mice (*Figure 4c*, *Figure 4—figure supplement 1a–d*; *Supplementary file 1*) and no changes in their mobility during the T-maze exploration (*Figure 4—figure supplement 1e, f*; nonTg: *n* = 10 animals; Tg: *n* = 5 animals; *Supplementary file 1*). Consistent with previous findings highlighting speed modulation of SOM-INs, PV-INs, and VIP-INs during locomotion (*Arriaga and Han, 2017*; *Francavilla et al., 2019*; *Turi et al., 2019*; *Luo et al., 2020*; *Geiller et al., 2020*), INs exhibited heightened activity during walk, but this pattern of activity remained unaffected when comparing nonTg and Tg mice (*Figure 4d, e*; nonTg: *n* = 9 animals; Tg: *n* = 8 animals; *Supplementary file 1*). However, during a decision-making spatial task in a T-maze (*Figure 4f*), both groups performed well (*Figure 4g*; nonTg: *n* = 9 animals; Tg: *n* = 6 animals; *Supplementary file 1*), yet activity of INs in Tg-mutants notably surged, particularly in the decision zone (D-zone) compared to the stem (*Figure 4h*; nonTg: *n* = 5 animals; Tg: *n* = 5 animals; *Supplementary file 1*), although the magnitude of change between the DZ and the stem remained unaltered (*Supplementary file 1*). Similarly, heightened activity of INs was observed in Tg mice during object exploration in the OFM (*Figure 4j*; nonTg:

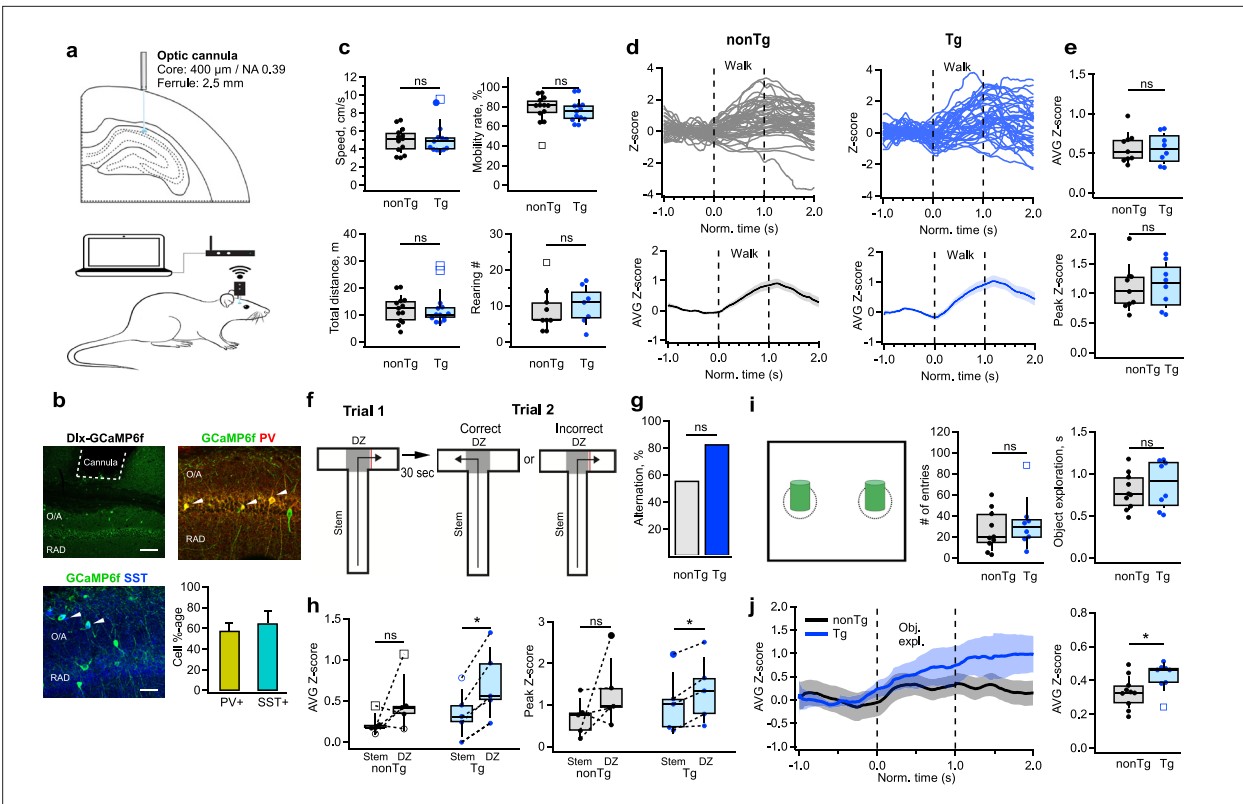

**Figure 4.** Enhanced activation of CA1 interneurons during decision-making and object exploration. (**a**) Schematic of cannula implantation and the experimental setup for wireless fiber photometry calcium imaging in freely behaving mice. (**b**) Representative confocal images of the CA1 hippocampal area showing the cannula track and GCaMPF6f expression in the CA1 parvalbumin- (PV-INs) or somatostatin-expressing (SOM-INs), with a bar graph summarizing the fraction of PV-INs and SOM-INs contributing to the photometry signal. White arrowheads depict examples of cell bodies expressing GCaMPF6f. Scale bars: top, 100 µm; bottom, 50 µm. (**c**) Summary plots showing the mean speed, mobility rate, total distance traveled (*n* = 13 animals per group), and the number of rearing episodes (nonTg: *n* = 9 animals; Tg: *n* = 7 animals) during the open field exploration in nonTg (black) and Tg (blue) animals. (**d**) Representative individual (top) and average (bottom) traces obtained from *stratum oriens/alveus* (O/A) INs in nonTg (left, black) and Tg (right, blue) mice during walking in the open field arena. (**e**) Summary plots showing the average (top) and peak (bottom) Z-scored values of calcium transients in O/A INs in nonTg (black) and Tg (blue) mice during walking (nonTg: *n* = 9 animals; Tg: *n* = 8 animals). (**f**) Schematic of the behavioral paradigm for the T-maze: correct-choice animals visit an alternative to the previously visited arm during the second trial. The decision zone (DZ) is illustrated with the shaded area. (**g**) Bar graph showing the percentage of alternation in T-maze test in nonTg and Tg animals (nonTg: *n* = 9 animals; Tg: *n* = 6 animals). (**h**) Summary plots showing the average (left) and peak (right) Z-scored values of calcium transients in O/A INs of nonTg (black) and Tg (blue) mice during exploration in the stem vs. DZ. Only mice making the correct choice (those that visited alternative to the previously visited arm) were included in this analysis. *p < 0.05 (nonTg: *n* = 5 animals; Tg: *n* = 5 animals). (**i**) Schematic of the arena used to examine the object-related modulation of neuronal activity (left), with summary plots showing the number of entries inside the object exploration zone (middle), and the average duration of object exploration episodes (right) (nonTg: *n* = 10 animals; Tg: *n* = 8 animals). (**j**) Representative average traces (left) obtained from O/A INs in nonTg (black) and Tg (blue) mice during object exploration and summary plot of the average Z-scored values of calcium transients recorded in interneurons during object exploration periods. *p < 0.05 (right, nonTg: *n* = 10 animals; Tg: *n* = 9 animals).

The online version of this article includes the following figure supplement(s) for figure 4:

**Figure supplement 1.** Unaltered activity of CA1 interneurons in home cage and open field and unchanged mobility in T-maze.

*n* = 10 animals; Tg: *n* = 9 animals; *Supplementary file 1*), despite no alterations in the number of entries in the objects' zones or in objects' exploration time (*Figure 4i*; nonTg: *n* = 10 animals; Tg: *n* = 8 animals; *Supplementary file 1*). Consequently, these data indicate that a mouse model carrying three familial AD mutations displays an early-age increase in the activity of hippocampal CA1 INs specifically during cognitive tasks associated with spatial decision-making and object encoding.

## Task-dependent modulation of CA1 PC activity in Tg mice

Ultimately, to explore whether changes in activity of CA1 INs influence the recruitment of PCs during behavioral tasks, we monitored the activation of CA1 PC using the *Camk2a*-promoter-driven GCaMP6f

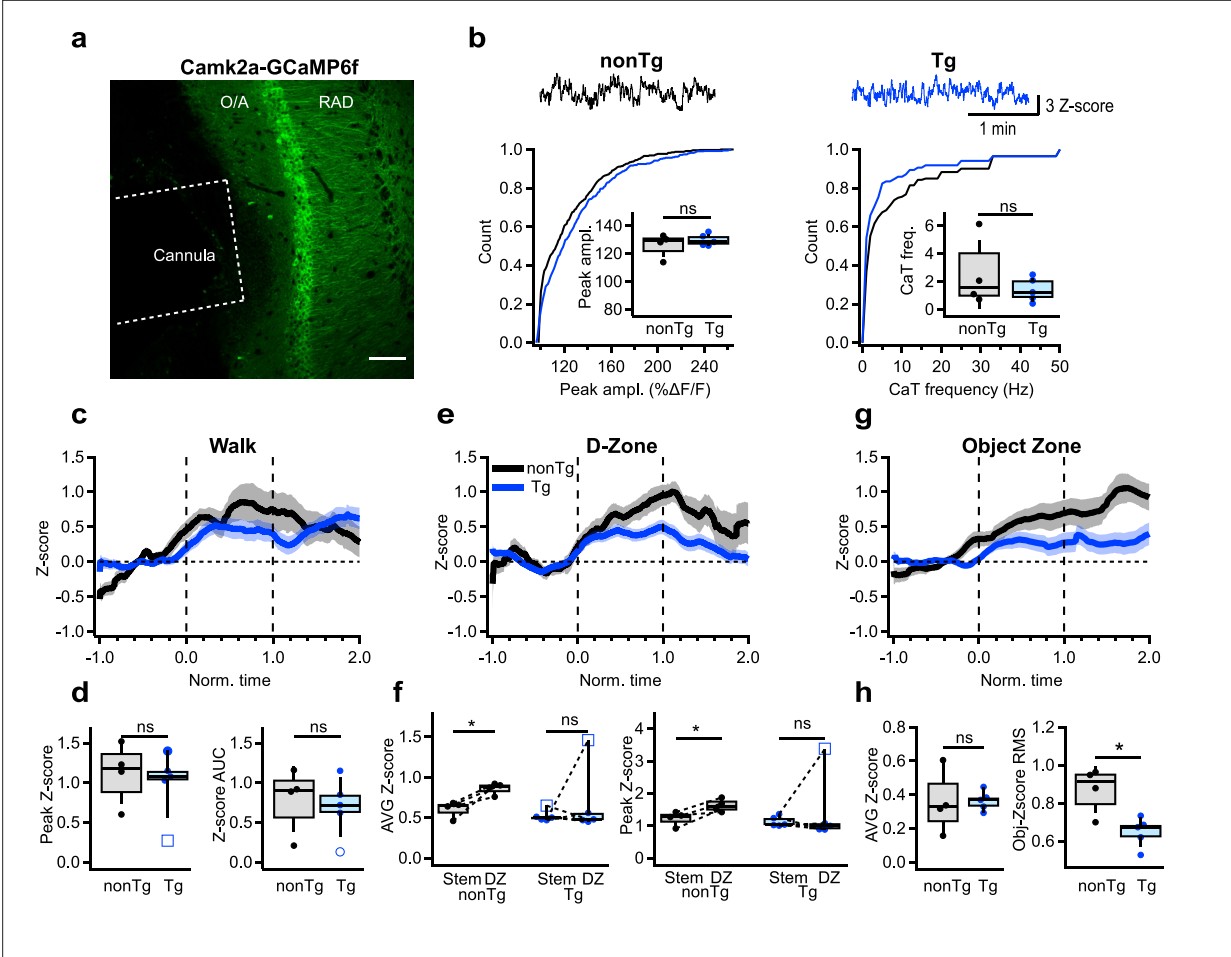

**Figure 5.** Altered activation of CA1 pyramidal cells during cognitive tasks. (**a**) Representative confocal image of the CA1 hippocampal area showing the cannula track and GCaMPF6f expression in the CA1 principal cells (PCs). Scale bar: 100 µm. (**b**) Representative calcium traces and summary plots showing the peak amplitude (left) and frequency (right) of calcium transients recorded in the CA1 PCs in nonTg (black) and Tg (blue) mice when animals were in their homecage (nonTg: *n* = 4 animals; Tg: *n* = 5 animals). Insets show summary boxplots for animal-wise mean comparison. (**c**) Representative average traces obtained from PCs in nonTg (black) and Tg (blue) mice during walking. The shaded areas show the SEM. (**d**) Summary plots showing the peak and area under the curve (AUC) for *Z*-scored values of calcium transients in PCs of nonTg (black) and Tg (blue) mice during walking (nonTg: *n* = 4 animals; Tg: *n* = 5 animals). (**e**) Representative average traces obtained from PCs in nonTg (black) and Tg (blue) mice during exploration in the D-zone of T-maze. The shaded areas show the SEM. (**f**) Summary plots showing the average (left) and peak (right) *Z*-scored values of calcium transients in PCs of nonTg (black) and Tg (blue) mice during exploration in the stem vs. DZ. *p <0.05 (nonTg: *n* = 4 animals; Tg: *n* = 5 animals). (**g**) Representative average traces obtained from PCs in nonTg (black) and Tg (blue) mice during object exploration. The shaded areas show the SEM. (**h**) Summary plots showing the AUC (left) and the root mean square (RMS) for *Z*-scored values of calcium transients in PCs of nonTg (black) and Tg (blue) mice during object exploration. *p < 0.05 (nonTg: *n* = 4 animals; Tg: *n* = 5 animals).

expression (*Figure 5a*; nonTg: *n* = 4 animals; Tg: *n* = 5 animals). Similar to INs, no alterations were observed in the baseline activity of PCs when mice were in their homecages; both the amplitude and frequency of calcium transients remained unaltered between Tg and nonTg animals (*Figure 5b*; *Supplementary file 1*). Moreover, there was no discernible change in PCs' activity between the two groups during locomotion in the OFM (*Figure 5c, d*; *Supplementary file 1*). However, during spatial decision-making in the T-maze, nonTg mice displayed a consistent increase in PC recruitment specifically within the D-zone. In contrast, Tg mice showed no change in PC activity in this area (*Figure 5e, f*; *Supplementary file 1*), consistent with the heightened activity observed in INs in D-zone for Tg mice (*Figure 4h*). Furthermore, during exploration of two different objects, while overall PC activity showed no difference between the two groups (*Figure 5h*; *Supplementary file 1*), the object modulation variance was reduced in Tg mice, revealing greater uniformity in signal during object sampling (*Figure 5g, h*; *Supplementary file 1*). Collectively, these findings reveal the task-dependent shifts in

PCs' recruitment in Tg mice, with unaltered activity during basal routine behavior and task-specific changes during spatial decision-making and object encoding.

## Discussion

In this study, we provide novel insights into the early alterations in the physiological properties of I-S3 cells in mice afflicted with AD neuropathology, along with the potential impact of these changes on the recruitement of the CA1 INs and PCs. Our findings indicate that at early stages in AD progression, APP/Aβ is detected intracellularly in I-S3 cells but they maintain their structural integrity. Notably, different subtypes of INs exhibit varying susceptibilities to AD-related pathologies (*Ali et al., 2023*), with CR-INs displaying anatomical resilience to pathological Aβ accumulation (*Shi et al., 2020*). This resilience in CR-INs, particularly in the early stages of AD, could be attributed to their ability to counteract Aβ-induced excitotoxicity by restricting cytosolic calcium elevations due to CR expression (*Mikkonen et al., 1999*; *Bezprozvanny and Mattson, 2008*; *Ali et al., 2023*). However, studies have demonstrated the toxicity of the intracellular accumulation of Aβ oligomers (*Li et al., 2009*; *Busche et al., 2012*) and their precursors (*Mondragón-Rodríguez et al., 2018*; *Takasugi et al., 2023*), highlighting how this may alter cellular processes and circuit function. For example, C99 fragments, which accumulate in 3-month-old 3xTg-AD mice (*Lauritzen et al., 2012*), are involved in the regulation of voltage-gated potassium channels of the Kv7 family (*Manville and Abbott, 2021*). Such a mechanism could modify the firing properties of affected neurons and may be at play in I-S3 cells of 3xTg-AD mice.

Notably, we observed significant alterations in the firing properties of I-S3 cells, characterized by a decrease in AP firing rate, potentially contributing to the heightened activity of O/A INs in specific tasks in freely behaving Tg mice. Functional changes in the neuronal circuits of the hippocampus have been widely recognized in animal models of AD (*Busche and Konnerth, 2016*; *Harris et al., 2020*). During the initial stages of the disease, there have been reports of hyperexcitability in the hippocampus and medial entorhinal cortex (MEC) (*Busche et al., 2008*; *Busche et al., 2012*; *Davis et al., 2014*; *Javonillo et al., 2021*; *Chen et al., 2023*), consistent with the increased susceptibility to epileptic seizures observed in both animal models and human patients (*Palop and Mucke, 2009*; *Cretin et al., 2016*). By providing disinhibition to principal excitatory cells of CA1, I-S3 cells are well positionned to coordinate hippocampal activity levels. Accordingly, in young adult 3xTg-AD mice, the altered firing of I-S3 cells might lead to abnormal disinhibition of PCs, thereby shifting the excitation/inhibition levels and affecting overall network activity. However, our observations revealed more intricate alterations in the output of I-S3 cells, characterized by slower individual APs and a decreased firing rate. The slower repolarization rate in I-S3 cells might arise from intrinsic changes involving Kv3.1 or Kv4.3 subunits expressed by I-S3 cells (*Guet-McCreight et al., 2016*), which have been suggested to occur during aging in I-S3 cells (*Francavilla et al., 2019*). Altered Kv3 properties reduce the excitability of cortical PV-INs and contribute to early AD-related network hyperexcitability (*Olah et al., 2022*). Moreover, increased intrinsic excitability exacerbated by enhanced synaptic inputs has been recently shown in MEC stellate cells in 3-month-old 3xTg-AD mice (*Chen et al., 2023*). Whether these simultaneous excitability changes in different cell types are initiated by the early intracellular Aβ accumulation or alterations in network activity remains to be determined, but collectively they can contribute to the onset of cognitive deficits. These early cell-type-specific excitability changes might also act as a compensatory mechanism to offset the altered connectivity motifs in order to normalize the network activity across different areas of the limbic system.

Furthermore, our in vivo data highlight task-specific alterations in the recruitment of CA1 O/A INs, which are the main target of I-S3 cells (*Tyan et al., 2014*), during spatial navigation and object encoding. If the behavioral task-dependent shifts in the recruitment of O/A INs are associated with alterations in I-S3 excitability, they should be most pronounced during tasks that notably elevate the I-S3 cells' activity. I-S3 cells are engaged in spatial decision-making, object coding and goal-directed behavior (*Magnin et al., 2019*; *Turi et al., 2019*; *Tamboli et al., 2024*). Consistent with the enhanced recruitment of these cells in the D-zone during spatial navigation and their positive modulation by objects (*Tamboli et al., 2024*), the activity of O/A INs was specifically enhanced during these behavioral patterns. Conversely, while I-S3 cells are modulated by animal running speed (*Turi et al., 2019*; *Luo et al., 2020*), no discernible alterations in the activity of O/A INs were noted during walking

periods. This is likely due to an overall lower animal speed during free walks in the open arena when compared to treadmill runs (*Turi et al., 2019*; *Luo et al., 2020*).

What could be the consequences on memory formation if there are changes in the recruitment of CA1 INs? The task-dependent increase in the activity of SOM-INs, the primary target of I-S3 cells, might lower dendritic excitability of CA1 PCs, impeding their input integration and burst firing (*Lovett-Barron et al., 2012*; *Royer et al., 2012*; *Tyan et al., 2014*; *Bilash et al., 2023*). This could potentially result in inadequate recruitment of CA1 PCs during decision-making or object sampling tasks. Indeed, our study reveals a decreased recruitment of CA1 PCs within the D-zone of the T-maze and their more uniform activation when exploring different objects. This aligns with increased inhibition and compromised integration of distinct sensory experiences (*Inayat et al., 2023*). Additionally, both in humans and in animal models of AD, early deficits in network oscillations have been documented (*Adler et al., 2003*; *Herrmann and Demiralp, 2005*; *van der Hiele et al., 2007*; *Czigler et al., 2008*; *Moretti et al., 2010*; *Scott et al., 2012*; *Goutagny et al., 2013*; *Schneider et al., 2014*; *Jones et al., 2019*). Theta oscillations, crucial for memory formation, are shaped by the recruitment of I-S3 cells (*Tyan et al., 2014*; *Luo et al., 2020*). Moreover, OLM cells, a primary target of I-S3 cells, play a significant role in generating theta oscillations (*Chatzikalymniou and Skinner, 2018*; *Mikulovic et al., 2018*; *Gu et al., 2020*). Changes in AP kinetics and firing rate of I-S3 cells could lead to aberrant activation of OLM cells (*Tyan et al., 2014*), potentially contributing to the early deficits in theta rhythms observed in human cases and AD animal models (*Goutagny and Krantic, 2013*; *Wirt et al., 2021*). This hypothesis could be further explored by imaging the activity levels of specific INs types, including OLMs, and their correlation with I-S3 activity during network oscillations and learning tasks in 3xTg-AD mice (*Luo et al., 2020*; *Geiller et al., 2020*). It is possible that alterations in interactions between I-S3 cells and their targets shift CA1 disinhibition necessary for sensory integration and decision-making (*Geiller et al., 2020*; *Bilash et al., 2023*) toward inhibition, impacting memory encoding dynamics and memory content, particularly in the early stages of AD progression. Understanding these deficits in specific circuit patterns tied to network states could pave the way to novel therapeutic approaches, focusing on early interventions and targeting precise circuit motifs to restore memory function.

# Materials and methods

## Mouse lines

All experiments were conducted in accordance with the Animal Protection Committee of Université Laval and the Canadian Council on Animal Care (Protocol # CHU-19045). This study involved VIP-eGFP [MMRRC strain #31009, STOCK Tg(Vip-EGFP)37Gsat, University of California, Davis, CA, USA], 3xTg-AD [MMRRC strain #034830-JAX, STOCK B6;129-Tg(APPSwe,tauP301L)1Lfa *Psen1^tm1Mpm*/Mmjax], and VIP-eGFP-3xTg (VIP-Tg) mouse lines. Triple-transgenic 3xTg-AD (Tg) mice (APPswe, PSIMI46V, tauP301L) (*Oddo et al., 2003*) were maintained in our animal facilities and backcrossed every seven to ten generations to C57BL6/129SvJ mice. VIP-Tg mice were generated through breeding homozygous Tg with heterozygous VIP-eGFP ones. The control groups comprised VIP-eGFP mice (VIP-nonTg) of the appropriate age (*Figures 1 and 2*) or non-transgenic littermates from the 3xTg-AD line (nonTg) (*Figures 3–5*). ARRIVE guidelines were applied in animal experiments in vivo. All experiments involved mice (P90–260) of both sexes, equally distributed across each experimental group. Animals were grouped based on age, and then randomization was done to ensure that each group is comparable. The sample size was determined in pilot experiments based on statistical power calculations.

## Behavior assays

General exploratory activity in the open field and the object recognition memory of VIP-Tg and VIP-nonTg mice were assessed using the automated VersaMax Animal Activity Monitoring System (AccuScan Instruments, Columbus, OH). The open field was a square arena (20 × 20 × 30.5 cm) made of plexiglass equipped with photocell beams for automatic detection of activity. On day 1 mice were placed in the OFM to explore it for 5 min (*Figure 1—figure supplement 1*). The same arena was used for the object-place task (OPT). On day 2, during the Sample phase, mice were placed in the OFM with two different sample objects (A + B; 4 × 4 cm each) for 5 min, after which they were returned to their home cage for 24 hr retention interval. On day 3, during the Test phase, mice re-entered the

arena with two objects, one was identical to the Sample phase and the other was novel (C + A; 4 × 4 cm each). As 3xTg-AD mice at this age do not typically show deficits in novel-object-recognition test (**Pairojana et al., 2021**), to add the complexity to this test, we altered the spatial location of the familiar object explored by the mouse. The results are expressed as an RI (%), defined as the ratio of the time spent with new object divided by the total exploration time for novel and familiar objects [RI $= T_N/(T_N + T_F)$]. An exploration time was also calculated for the time spent with the two objects in the Test phase (ET$_T$ $= T_N + T_F$). Control mice show preference for a novel object in this test (**Francavilla et al., 2020**).

To investigate object modulation of neuronal activity, another group of mice were positioned in a rectangular arena (36 × 29 × 21.5 cm) made of opaque plastic equipped with video camera for automatic detection of activity. The animal activity (immobility, locomotion, and rearing periods) was monitored using the AnyMaze software. Animal tracking was performed using the ToxTrac software v2.96 (**Rodriguez et al., 2018**) to assess speed, mobility rate and distance traveled. Two identical objects (A + A; 4 × 4 cm each) were used for object modulation assessment.

T-maze was performed in a T-shaped plexiglass maze (80 × 68 × 15 cm). A single mouse was placed at the beginning of the maze and restricted manually to the selected arm for 30 s using a cardboard door. The mouse was then repositioned at the beginning and left to explore for a total trial duration of 5 min. To assess spatial memory, selected arm alternation was measured. To measure calcium activity during decision-making periods, a decision zone was defined as the square area located at the intersection of the maze (**Figure 4f**). Between animals the arena and objects were cleaned with 70% ethanol.

We excluded from the analysis the animals that refused to explore the T-maze and spent all their time in the stem corner, or refused to explore the objects and stayed in the OFM corner. These exclusions applied to both nonTg ($n$ = 6 mice) and Tg ($n$ = 5 mice) groups, indicating that low exploratory activity is not necessarily linked to AD-related mutations. During the T-maze test, we also observed several animals that made incorrect choices (four out of nine nonTg and one out of six Tg mice). However, due to the low number of animals making incorrect choices, we were unable to form a separate group for analysis based on incorrect choices.

## Slice preparation and patch-clamp recordings

Transversal hippocampal slices (thickness, 300 μm) were prepared from VIP-nonTg, VIP-Tg, nonTg, or Tg mice of either sex. The animals were anesthetized with ketamine–xylazine mixture (10/100 mg/ml), transcardially perfused with 20 ml of ice-cold sucrose-based cutting solution (containing the following in mM: 250 sucrose, 2 KCl, 1.25 NaH$_2$PO$_4$, 26 NaHCO$_3$, 7 MgSO$_4$, 0.5 CaCl$_2$, and 10 glucose, pH 7.4, 330–340 mOsm/l) and decapitated (**David and Topolnik, 2017**). Slices were cut in the cutting solution using a vibratome (VT1000S; Leica Microsystems or Microm; Fisher Scientific), and transferred to a heated (37.5°C) oxygenated recovery solution containing the following (in mM): 124 NaCl, 2.5 KCl, 1.25 NaH$_2$PO$_4$, 26 NaHCO$_3$, 3 MgSO$_4$, 1 CaCl$_2$, and 10 glucose; pH 7.4; 300 mOsm/l, to recover for at least 45 min. During experiments, slices were continuously perfused (2 ml/min) with standard artificial cerebrospinal fluid containing the following (in mM): 124 NaCl, 2.5 KCl, 1.25 NaH$_2$PO$_4$, 26 NaHCO$_3$, 2 MgSO$_4$, 2CaCl$_2$, and 10 glucose, pH 7.4 saturated with 95% O$_2$ and 5% CO$_2$ at near physiological temperature (30–33°C). VIP-expressing INs located in RAD and PYR were visually identified as eGFP-expressing cells under an epifluorescence microscope with blue light (filter set: 450–490 nm). All electrophysiological recordings were carried out using a ×40 water-immersion objective. A Flaming/Brown micropipette puller (Sutter Instrument Co) was used to make patch pipettes (3.5–6 MΩ). Whole-cell patch-clamp recordings from VIP or O/A INs were performed in voltage- or current-clamp mode. Pipettes filled with K$^+$-based (for current-clamp) or Cs$^+$-based (for voltage-clamp) solution: 130 KMeSO$_4$/CsMeSO$_4$, 2 MgCl$_2$, 10 diNaphosphocreatine, 10 HEPES, 4 ATP-Tris, 0.4 GTP-Tris, and 0.3% biocytin (Sigma), pH 7.2–7.3, 280–290 mOsm/l. QX-314 was included in the intracellular solution in voltage-clamp recordings. Passive and active membrane properties were analyzed in current clamp mode. Passive membrane properties (resting membrane potential, input resistance, and membrane capacitance) were obtained during the first minute after membrane rupture. The cell firing was recorded by subjecting cells to multiple current step injections of varying amplitudes (20–200 pA). Voltage-clamp recordings were performed to analyze the excitatory and inhibitory drives received by VIP or O/A INs. For recordings of sEPSCs, the holding potential was fixed at −70 mV. sIPSCs were

recorded at 0 mV. Data acquisition (filtered at 2–3 kHz and digitized at 10 kHz; Digidata 1440, Molecular Devices, CA, USA) was performed using the Multiclamp 700B amplifier and the Clampex 10.9 software (Molecular Devices). Series resistance (in voltage-clamp) or bridge balance (in current-clamp) was monitored throughout the experiment. Cells that had changes in series resistance or bridge balance (>15%), a resting membrane potential more positive than −45 mV or showed an increase in holding current (>−30 pA) during recording were discarded.

## Electrophysiological data analysis

Analysis of electrophysiological recordings was performed using Clampfit 10.9 (Molecular Devices). For the analysis of the AP properties, the first AP appearing at the rheobase current pulse within a 50-ms time window was analyzed. The AP threshold, amplitude, half-width, depolarization and repolarization rates, and the AP area were detected as previously described (*Francavilla et al., 2020*). The number of APs was assessed at the current pulse of +140–150 pA.

For the analysis of spontaneous synaptic currents, a minimum of 100 events (for EPSCs) and 200 events (for IPSCs) were sampled per cell over a 10-min period using an automated template search algorithm in Clampfit. All events were counted for frequency analysis.

## Anatomical reconstruction and immunohistochemistry

For post hoc reconstruction, during whole-cell recordings neurons were filled with biocytin (Sigma). Slices containing the recorded cells were fixed overnight with 4% paraformaldehyde (PFA) at 4°C. To reveal biocytin, the slices were permeabilized with 0.3% Triton X-100 and incubated at 4°C with streptavidin-conjugated Alexa-488 (1:1000) in Trizma-buffer overnight. Z-stacks of biocytin-filled cells were acquired with a 1μm step using a ×20 oil-immersion objective and Leica SP5 confocal microscope. Confocal stacks were merged for detailed reconstruction in Neurolucida 8.26.2 (MBF Bioscience). The I-S3 cells were identified based on the presence of axonal projections in the O/A (*Acsády et al., 1996*; *Tyan et al., 2014*). The quantitative analysis of soma area and dendritic morphology was performed in Neurolucida (*Figure 2—figure supplement 2*). Sholl analysis was performed in radial coordinates, using a 50-μm step size with $r = 0$ centered on the cell soma.

All immunohistochemical tests were performed on free-floating hippocampal sections (50–70 μm thick) obtained with Leica VT1000S or PELCO EasySlicer vibratomes from mice perfused with 4% PFA. Sections were permeabilized with 0.25% Triton X-100 in PBS (Phosphate-Buffered Saline) and incubated overnight at 4°C with primary antibodies followed by the secondary antibodies. The following primary antibodies were used in this study: goat anti-CR (1:1000; Santa Cruz, #sc-11644), chicken anti-GFP (1:1000; Aves, #GFP-1020), mouse anti-β-amyloid, 1–16 (6E10) (1:3000; Covance, #SIG-39320), anti-human PHF-tau (1:1000; Thermo Scientific, #MN1020), and mouse anti-VGAT (1:500; Synaptic Systems, #131011). Control immunohistochemical tests were performed by omitting the primary antibodies and incubating the sections in the mixture of secondary antibodies. Confocal images were acquired sequentially using a Leica TCS SP5 imaging system coupled with a 488-nm argon, a 543-nm HeNe, and a 633-nm HeNe lasers. Cell counting was performed using stereological analysis within randomly selected regions of interest (500 × 700 μm) in the CA1 area. To sample the same hippocampal area along the septotemporal axis, only middle sections were analyzed. The cell counting was performed in 4–8 sections per animal from 3 to 4 different animals. For inhibitory axonal boutons density analysis, images of the CA1 O/A layer were acquired and the density of VGAT+ or VGAT+/CR+ boutons was measured using ImageJ (v1.53). VGAT+ puncta (0.5–0.8 μm) were counted automatically using the 'Analyze particles' function with watershed segmentation within a 104 μm$^2$ area in the O/A in a single focal plan confocal image (2048 × 2048 pxl) at the same focal depth for all sections.

## Stereotaxic injection and cannula implantation

Before stereotaxic surgery, nonTg or Tg mice were administered with buprenorphine slow release (0.1 mg/kg; intraperitoneal (i.p.)) and lidocaine–bupivacaine (0.1 ml; local application at the site of incision). Mice were anesthetized with isoflurane and fixed in a stereotaxic frame (RWD). After incision, a hole was drilled in the skull over the site of injection in the right hemisphere. To express GCaMP6f in the CA1 region of the hippocampus, viral vectors were injected at the following coordinates: AP −2.2, ML −2, DV, −1.3 mm using a micro-pipette, which was attached to a microprocessor-controlled

nanoliter injector (Nanoliter 2000; World Precision Instruments). AAV.mDlx.GCaMP6f (Addgene, #83899) was used to target inhibitory INs and AAV1-*Camk2a*-GCaMP6f-WPRE (Penn Vector Core) was used to target primarily pyramidal neurons. The total volume of 100 nl was injected at the rate of 10 nl/s. After injection, the pipette was kept at the site for 10 min and then withdrawn slowly to prevent backflow of the virus. For calcium imaging using wireless fiber photometry, the TeleFipho cannula (core 400 mm, NA 0.39, cladding 425 mm, ferrule 2.5 mm) was implanted at the same stereo-taxic coordinates. Cannula was fixed to the skull with quick adhesive cement (C&B-Metabond), and the exposed skull was covered with dental cement mixed with carbon powder. Mice were injected with lactate ringer solution (0.1 ml/10 mg) and carprofen (0.1 ml/10 mg) post-surgery and for 1 day after surgery. Animal weight was monitored for 5 days post-surgery. Animals were allowed to recover for 2 weeks before the beginning of handling and calcium imaging experiments. After fiber photom-etry experiments, mice were perfused with 4% PFA and brain were sectioned (70 µm) on the next day. Sections containing the cannula track were mounted and confocal images were collected using the oil-immersion ×20 (NA 0.8) objective and a laser-scanning Leica SP5 confocal microscope, and the expression of GCaMP6f as well as the cannula position was confirmed. Animals with imprecise cannula location were discarded from the analysis.

## Wireless fiber photometry for in vivo calcium imaging

Calcium imaging was performed using the TeleFipho wireless fiber photometry device (Amuza Inc). The device calibration was done as previously described (*Amalyan et al., 2022*). Starting from 2 weeks post-injection of viral vectors, mice were exposed to daily handling for 4–5 days and habituated with a dummy head-stage attached to the implanted cannula for 2 days in their homecage before the start of behavioral studies. Following habituation, the TeleFipho head-stage was attached to the implanted cannula and mice were placed in their homecage or an arena where they were allowed to explore for 5 min. During this period, fiber photometry calcium signal was acquired wirelessly by the TeleFiR receiver (Amuza Inc) at a sampling rate of 100 Hz. The calcium signal was quantified as the *Z*-score or ΔF/F, calculated in MATLAB with the following equations:

$$Z - \text{score} = \frac{\left(F_{\text{corrected}} - \mu\right)}{\sigma}$$

where $F_{\text{corrected}}$ is the de-trended signal, $\mu$ is the mean, and $\sigma$ is the standard deviation. The mean, maximum, and area under the curve of the *Z*-score during a particular behavioral state was obtained using a MATLAB script detecting the dynamic threshold for calculation of *Z*-score.

$$\frac{\Delta F}{F} = \frac{\left(F_{\text{corrected}} - F_0\right)}{\text{abs}\left(F_0\right)}$$

where $F_0$ is the minimum value in the $F_{\text{corrected}}$ trace.

The *Z*-scores were obtained using scripts written in MATLAB. The representative average traces of *Z*-scores for the period before, during, and after the walk or exploration of D-zones or objects (*Figures 4 and 5*) were extracted using code written in MATLAB. First, to present the activity dynamics during walk/exploration bouts of variable durations using a representative curve, the event durations were normalized. Second, the time series of each event were converted to proportions that is, each time point in the time series was divided by the total duration of the event. Hence, the values on the *x*-axis can be considered as a fraction of the total event time where '0' and '1' means the start and end of the event, respectively. The object modulation variance (*Figure 5h*) was measured as the root mean square of mean neuronal activity in the object zone.

## Statistics

All statistical analysis was conducted in Sigma Plot 11.0, Clampfit 10.6, and Igor Pro. The datasets were first tested for normality with a Shapiro–Wilcoxon or Runs test. If data were normally distributed, standard parametric statistics were used: unpaired *t* tests for comparisons of two groups and two-way analysis of variance for comparisons of multiple groups followed by Brown–Forsythe test. If data were not normally distributed, non-parametric Mann–Whitney or Wilcoxon rank tests were used for comparisons of two groups. The 'p' values <0.05 were considered significant.

For statistical analysis of electrophysiological recordings, we based our findings analyzing the data with linear mixed model (LMM), modeling animal as a random effect and genotype as fixed effect. We used this statistical analysis because we considered the number of mice as independent replicates and the number of cells in each mouse as repeated measures. LMM analysis was performed using IBM SPSS V29.0.0 software.

The data are presented using box plots and violon plots and show data distribution. Error bars represent the standard deviation, and the statistical results are presented as *p < 0.05, **p < 0.01, ***p < 0.001.

## Additional information

### Funding

| Funder | Grant reference number | Author |
| --- | --- | --- |
| Canadian Institutes of Health Research | MOP-137072 | Lisa Topolnik |
| Canadian Institutes of Health Research | MOP-142447 | Lisa Topolnik |
| Natural Sciences and Engineering Research Council of Canada | 342292 | Lisa Topolnik |

The funders had no role in study design, data collection, and interpretation, or the decision to submit the work for publication.

### Author contributions

Felix Michaud, Formal analysis, Investigation, Methodology, Writing – original draft, Writing – review and editing; Ruggiero Francavilla, Formal analysis, Investigation, Visualization, Methodology, Writing – original draft, Writing – review and editing; Dimitry Topolnik, Formal analysis, Investigation, Methodology; Parisa Iloun, Suhel Tamboli, Formal analysis; Frederic Calon, Resources, Writing – review and editing; Lisa Topolnik, Conceptualization, Resources, Data curation, Software, Formal analysis, Supervision, Funding acquisition, Validation, Investigation, Visualization, Methodology, Writing – original draft, Project administration, Writing – review and editing

### Author ORCIDs

Lisa Topolnik https://orcid.org/0000-0003-4236-5473

### Ethics

All experimental protocols using laboratory animals were approved by the Animal Protection Committee of Université Laval (CPAUL) and performed following the guidelines of CPAUL and the Canadian Council of Animal Care (CCAC) (Protocol #CHU-19045).

Reviewer #1 (Public Review): https://doi.org/10.7554/eLife.95412.3.sa1
Author response https://doi.org/10.7554/eLife.95412.3.sa2

## Additional files

### Supplementary files

• Supplementary file 1. Summary table of statistical analysis conducted throughout the study. Notes: (1) The reported *n* corresponds to the number of animals, cells (underlined), or slices (italic). (2) Animals of both sexes with equal distribution were used throughout the study.

• MDAR checklist

### Data availability

All data generated or analyzed during this study are included in the manuscript and supporting files.

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
