## [Editor Report · eLife assessment]

This study describes **fundamental** findings related to early disruptions in disinhibitory modulation exerted by VIP+ interneurons, in CA1 in a transgenic model of Alzheimer's disease pathology. The authors provide a **compelling** analysis at the cellular, synaptic, network, and behavioral levels on how these changes correlate and might be related to behavioral impairments during these early stages of AD pathology.

---

## [Referee Report · Reviewer #1 (Public Review)]

Summary:

The work in the manuscript utilized patch-clamp techniques to explore the electrophysiological characteristics of VIP interneurons in the early stages of AD using the 3xTg mouse model. The study revealed that VIP interneurons exhibited prolonged action potentials and reduced firing rates. These changes could not be attributed to modifications in input signals or morphological transformations. The authors attributed aberrant VIP activity to the accumulation of beta-amyloid in those interneurons.

The decreased frequency of VIP inhibitory events were associated with no observed changes in excitatory drive to these interneurons. Consequently, heightened activity in the general population of CA1 interneurons was observed during a decision-making task and an object recognition test. In light of these findings, the authors concluded that the altered firing patterns of VIP interneurons may initiate early-stage dysfunction in hippocampal CA1 circuits, potentially influencing the progression of AD pathology.

Strengths:

Overall the work is novel and moves the field of Alzheimer's disease forward in a significant way. The manuscript reports a novel concept of aberrant activity in VIP interneurons during the early stages of AD thus contributing to dysfunctions of the CA1 microcircuit. This results in enhancement of the inhibitory tone on the primary cells of CA1. Thus, the disinhibition by VIP interneurons of Principal Cells is dampened. The manuscript was skillfully composed, the study was of strong scientific rigor featuring well-designed experiments. Necessary controls were present. Both sexes were included.

Major limitations were not adequately addressed in the revised manuscript

(1) The authors attributed aberrant circuit activity to accumulation of "Abeta intracellularly" inside IS-3 cells. That is problematic. 6E10 antibody recognizes amyloid plaques in addition to Amyloid Precursor Protein (APP) as well as the C99 fragment. There are no plaques at the ages 3xTg mice were examined. Lack of plaques was addressed in revised manuscript. The staining shown in Fig. 1a is of APP/C99 inside neurons, not abeta accumulations in neurons. At the ages of 3-6 months, 3xTg mice start producing and releasing extracellular abeta oligomers and potentially tau oligomers as well (Takeda et al., 2013 PMID: 23640054; Takeda et al., 2015 PMID: 26458742 and others). Emerging literature suggests that extracellular not intracellular abeta and tau oligomers disrupt circuit function. Thus, a more likely explanation of extracellular abeta and tau oligomers disrupting the activity of VIP neurons is plausible. Presence of intracellular abeta is currently controversial in the field and needs to be discussed as such. Some of the references added in the revised version of the manuscript are erroneously cited. The authors provide no original data in support of "intracellular" abeta.

(2) Authors suggest that their animals do not exhibit loss of synaptic connections and show Fig. 3d in support of that suggestion. However, imaging with confocal microscopy of 70 micron thick sections would not allow resolution of pre- and post-synaptic terminals. More sensitive measures such as electron microscopy or array tomography are the appropriate techniques to pursue. It is important for the authors to either remove that data from the manuscript or address/discuss the limitations of their technique in the discussion section. There is a possibility of loss of synaptic connections in their mouse model at the ages examined. Discussion of that possibility and of the limitations of the methodology used is missing.

---

## [Author Response]

The following is the authors’ response to the original reviews.

**Public Reviews:**

**Reviewer #1 (Public Review):**
Strengths:Overall the work is novel and moves the field of Alzheimer's disease forward in a significant way. The manuscript reports a novel concept of aberrant activity in VIP interneurons during the early stages of AD thus contributing to dysfunctions of the CA1 microcircuit. This results in the enhancement of the inhibitory tone on the primary cells of CA1. Thus, the disinhibition by VIP interneurons of Principal Cells is dampened. The manuscript was skillfully composed, and the study was of strong scientific rigor featuring well-designed experiments. Necessary controls were present. Both sexes were included.

We express our gratitude to the reviewer for their keen appreciation of our efforts and their enthusiasm for the outcomes of this research.

Limitations:(1) The authors attributed aberrant circuit activity to the accumulation of "Abeta intracellularly" inside IS-3 cells. That is problematic. 6E10 antibody recognizes amyloid plaques in addition to Amyloid Precursor Protein (APP) as well as the C99 fragment. There are no plaques at the ages 3xTg mice were examined. Thus, the staining shown in Figure 1a is of APP/C99 inside neurons, not abeta accumulations in neurons. At the ages of 3-6 months, 3xTg starts producing abeta oligomers and potentially tau oligomers as well (Takeda et al., 2013 PMID: 23640054; Takeda et al., 2015 PMID: 26458742 and others). Emerging literature suggests that abeta and tau oligomers disrupt circuit function. Thus, a more likely explanation of abeta and tau oligomers disrupting the activity of VIP neurons is plausible.

The Reviewer correctly points out that 3xTg-AD mice typically do not exhibit plaques before 6 months of age, with limited amounts even up to 12 months, particularly in the hippocampus. To the best of our knowledge, the 6E10 antibody binds to an epitope in APP (682-687) that is also present in the Abeta (3-8) peptide. Consequently, 6E10 detects full-length APP, α-APP (soluble alpha-secretase-cleaved APP), and Abeta (LaFerla et al., 2007). Nonetheless, we concur with the Reviewer's observation that the detected signal includes Abeta oligomers and the C99 fragment, which is currently considered an early marker of AD pathology (Takasugi et al., 2023; Tanuma et al., 2023). Studies have demonstrated intracellular accumulation of C99 in 3-month-old 3xTg mice (Lauritzen et al., 2012), and its binding to the Kv7 potassium channel family, which results in inhibiting their activity (Manville and Abbott, 2021). If a similar mechanism operates in IS-3 cells, it could explain the changes in their firing properties observed in our study. Consequently, we have revised the manuscript to include this crucial information in both the Results and Discussion sections.

(2) Authors suggest that their animals do not exhibit loss of synaptic connections and show Figure 3d in support of that suggestion. However, imaging with confocal microscopy of 70micron thick sections would not allow the resolution of pre- and post-synaptic terminals. More sensitive measures such as electron microscopy or array tomography are the appropriate techniques to pursue. It is important for the authors to either remove that data from the manuscript or address the limitations of their technique in the discussion section. There is a possibility of loss of synaptic connections in their mouse model at the ages examined.

We appreciate the Reviewer’s perspective on the techniques used for imaging synaptic connections. While we acknowledge the limitations of confocal microscopy for resolving pre- and post-synaptic structures in thick sections, we respectfully disagree regarding the exclusive suitability of electron microscopy (EM). Our approach involved confocal 3D image acquisition using a 63x objective at 0.2 um lateral resolution and 0.25 Z-step, providing valuable quantitative insights into synaptic bouton density. Despite the challenges posed by thick sections, this method together with automatic analysis allows for careful quantification. Although EM offers unparalleled resolution, it presents challenges in quantification. We have included the important details regarding image acquisition and analysis in the revised manuscript.

**Reviewer #2 (Public Review):**
Summary:The submitted manuscript by Michaud and Francavilla et al., is a very interesting study describing early disruptions in the disinhibitory modulation exerted by VIP+ interneurons in CA1, in a triple transgenic model of Alzheimer's disease. They provide a comprehensive analysis at the cellular, synaptic, network, and behavioral level on how these changes correlate and might be related to behavioral impairments during these early stages of the disease.Main findings:- 3xTg mice show early Aß accumulation in VIP-positive interneurons.- 3xTg mice show deficits in a spatially modified version of the novel object recognition test. - 3xTg mice VIP cells present slower action potentials and diminished firing frequency upon current injection.- 3xTg mice show diminished spontaneous IPSC frequency with slower kinetics in Oriens / Alveus interneurons.- 3xTg mice show increased O/A interneuron activity during specific behavioral conditions. - 3xTg mice show decreased pyramidal cell activity during specific behavioral conditions.Strengths:This study is very important for understanding the pathophysiology of Alzheimer´s disease and the crucial role of interneurons in the hippocampus in healthy and pathological conditions.

We are thankful to the reviewer for their insightful recognition of our efforts and their enthusiasm for the results of this research.

Weaknesses:Although results nicely suggest that deficits in VIP physiological properties are related to the differences in network activity, there is no demonstration of causality.

We completely agree with the reviewer's observation regarding the lack of demonstration of causality in our results. Investigating causality in the relationship between deficits in VIP physiological properties and differences in network activity is indeed a crucial aspect of this project. However, achieving this goal will require a significant amount of time and dedicated manipulations in a new mouse model (VIP-Cre-3xTg). We appreciate the importance of this line of investigation and consider it as a priority for our future research endeavors.

**Recommendations for the authors:**

**Reviewer #1 (Recommendations for the authors):**
Limitations:(1) The authors should describe their model and state the age at which these mice start depositing amyloid plaques and neurofibrillary tangles. Readers might not be familiar with this model. It is also important to mention that circuit disruptions are assessed prior to plaque and tangle formation.

We have included a detailed description of the 3xTg-AD mouse model in the Introduction section, including information on the age at which amyloid plaques and neurofibrillary tangles begin to appear. Additionally, we have clarified that circuit disruptions were assessed before the formation of plaques and tangles. These details have been added to both the Introduction and the Results sections to ensure clarity for readers unfamiliar with the model.

(2) Ns are presented in Supplemental Table 1. Units are presented in a note to Supplementary Table 1. It would be advisable to specify Ns and units as the data is being presented in the results section or figure legends for easy access.

We have now included the Ns (sample sizes), specifying the number of cells or sections and the number of experimental animals, directly within the Results section and in the figure legends. This ensures that readers have immediate access to this information without needing to refer to the supplementary materials.

(3) Several typos require correction:a. "mamory" - Line 22, page 5.b. The term "Interneurons" is abbreviated as both "INs" and "IN" throughout the manuscript. The author should consistently choose one abbreviation.

We have corrected the typo "mamory" to "memory" on line 22, page 5. Additionally, we have standardized the abbreviation for "Interneurons" to "INs" throughout the manuscript for consistency.

(4) Note 2 in Supplementary Table 1 states that animals of both sexes with equal distribution were used throughout the study. It would be best for the reader to assess the data distribution based on sex. Thus, it is advisable for the authors to depict male and female data points as distinct symbols throughout the figures.

Unfortunately, we do not have detailed sex-disaggregated data for all datasets, which limits our ability to depict male and female data points separately across all figures. Therefore, we have opted to pool data from both sexes for a more comprehensive analysis. We believe this approach maintains the robustness of our findings.

**Reviewer #2 (Recommendations for the authors):**
Major Points:- To keep the logical line of reasoning and to be able to interpret the results, it would be important to use the same metrics when comparing the population activity of O/A interneurons and principal cells in the different behavioral conditions.

We have revised Figures 4 and 5 to enhance the coherence in data presentation. This includes using consistent metrics for comparing the population activity of both O/A interneurons and principal cells across different behavioral conditions. These changes ensure a clearer and more logical interpretation of the results.

- Although results nicely suggest that deficits in VIP physiological properties are related to the differences in network activity, there is no demonstration of causality. Would it be possible to test if manipulating VIP neurons one could obtain such specific results? Alternatively, it could be discussed more in detail how the decrease in disinhibition could lead to the changes in network activity demonstrated here.

We agree with the reviewer that establishing causality between VIP neuron deficits and changes in network activity would be very important. However, demonstrating causality would require a new line of investigation, involving the use of specific mouse models to selectively manipulate VIP neurons. This is an exciting direction that we plan to prioritize in our future research. For this study, we have included a discussion on the potential mechanisms by which decreased disinhibition might lead to the observed changes in network activity. Specifically, we propose that in young adult 3xTg-AD mice, the altered firing of I-S3 cells may lead to enhanced inhibition of principal cells. This could shift the excitation/inhibition balance, input integration and firing output of principal cells thereby impacting overall network activity. These points are discussed in detail in the revised Discussion section.

- On the same lines the correlations showed in the manuscript, would be more robust if there was an in vivo demonstration that 3xTg mice indeed show decreased activity in vivo. The same experiments could also clarify if VIP cells in control animals are more active at the time of decision-making and during object exploration as suggested in the manuscript.

Thank you for your comment. In response to the point raised, we would like to highlight that we have recently documented the increased activity of VIP-INs in the D-zone of the T-maze and during object exploration in a study published in Cell Reports (Tamboli et al., 2024). This publication is now referenced in our manuscript to support our findings. Regarding the in vivo activity of 3xTg mice, our observations indicated no significant differences in major behavioral patterns such as locomotion, rearing, and exploration of the T-maze when comparing Tg and non-Tg mice. These findings are presented in detail in Figure 4c and Supplementary Fig. 5. We believe these data support the robustness of our correlations by demonstrating that the overall behavioral activity of 3xTg mice is comparable to that of non-transgenic controls, thus focusing attention on the specific roles of VIP-INs in early prodromal state of AD pathology.

Minor Points:- Figure 1c: Heading of VIP-Tg should have capital letters.Thank you for pointing that out. We have corrected the heading to "VIP-Tg" with capital letters in Figure 1c.- Figure 1d: The finding that no change was observed in the percentage of VIP+/CR+ is based on three animals and 3-4 slices per mouse. However, the result of VIP+CR+ in tg-mice has an outlier that might bias the results. I would suggest increasing the number of animals to confirm these results.

Thank you for your insightful suggestion. We addressed the potential impact of the outlier in the VIP+/CR+ cell density analysis by recalculating the results after removing the outlier using the interquartile range method. This reanalysis revealed a statistically significant difference in the VIP+/CR+ cell density between non-Tg and Tg mice, which we have now detailed in the Results section. Despite this, we have chosen to retain the outlier in our final presentation to accurately represent the biological variability observed in our sample. We agree that increasing the number of animals would further validate these findings and will consider this in future studies.

- Figure 3d: Would it be possible to identify the recorded interneurons? Is it expected that most of those are OLM cells?

Thank you for your question. We were unable to fully recover all recorded cells using biocytin staining. However, for those cells with preserved axonal structures, we identified both OLM and bistratified cells, which are the primary targets of I-S3 cells. We have now included this information in the Results section to clarify the types of interneurons identified.

- Figure 3: Why quantify VGat terminals instead of quantification of VIP-GFP terminals? Combined with the Calretinine labeling it would be more useful to indicate that no changes were observed at the morphological bouton level specifically in disinhibitory interneurons. Please also describe which imageJ plugin was used for the quantification.

Thank you for your question. Our primary objective was to quantify the synaptic terminals of CR+ INs in the CA1 O/A region, which are predominantly formed by I-S3 cells. Therefore, VGaT and CR co-localization was used to guide this analysis. GFP expression in axonal boutons can sometimes be inconsistent and less reliable for precise quantification. For this analysis, we utilized the “Analyze Particles” function in ImageJ, combined with watershed segmentation, which is now specified in the Methods section.

- Figure 4g: How was the statistical test performed? If data was averaged across mice, please add error bars and data points in the figure.

Thank you for your question. To compare the alternation percentage between non-Tg and Tg mice, we used Fisher’s Exact test as detailed in Supplementary Table 1. In this analysis, we considered each animal's choice individually, comparing the preference for correct versus incorrect choices between the two groups. Since Fisher’s Exact test is designed for analyzing qualitative data rather than quantitative data, averaging across mice was not applicable, and therefore, we did not include error bars or data points in the figure.

- Figure 4h: To conclude that the increase in activity is larger in the 3xTg mice, there should be a statistical comparison for the magnitude of change between the decision and the stem zone for control and 3xTg mice. To show that there is no significant difference in this measurement in the control mice is insufficient.

Thank you for your suggestion. We performed a statistical comparison of the magnitude of change in activity between the stem zone and the D-zone for non-Tg and 3xTg mice, as recommended. Our analysis showed no significant difference in this magnitude of change between the two genotypes. These results have now been included in the Results section. However, we would like to highlight an important finding regarding the nature of these changes. In the 3xTg mice, there was a consistent increase in the activity of O/A INs when entering the Dzone. In contrast, non-Tg mice displayed a range of responses, including both increases and decreases in activity. This indicates a higher reliability in the firing of O/A INs in the D-zone of 3xTg mice. Our recent study suggests that VIP-INs are particularly active in the D-zone (Tamboli et al., 2024). Therefore, the absence or reduced input from VIP-INs in 3xTg mice may lead to the observed higher engagement of O/A INs in this zone. We believe this observation is crucial for understanding the differential yet nuanced changes in neural dynamics in these mice.

- In the methods, it is stated that there was a pre-selection of animals depending on learning performance. Would it be possible to also show the data from animals that did not properly learn? Alternatively, it would be useful to plot the correlation between performance in this test and the difference between activity in the stem and the decision-making zone. The reason to ask for this is that there is a trend for control animals to show reduced alternations (50 vs 80%, although not significant, it is a big difference). Considering that there is also a trend in control animals to show increased activity in the decision-making zone, it would be important to confirm that this is not only due to differences in performance. The current statistical procedure does not allow discarding this.

In this study, we excluded from the analysis the animals that refused to explore the T-maze and spent all their time in the stem corner, or refused to explore the objects and stayed in the open field maze (OFM) corner. These exclusions applied to both non-Tg (n = 6) and Tg (n = 5) groups, indicating that low exploratory activity is not necessarily linked to AD-related mutations. During the T-maze test, we also observed several animals that made incorrect choices (4 out of 9 non-Tg and 1 out of 6 Tg mice). However, due to the low number of animals making incorrect choices, we were unable to form a separate group for analysis based on incorrect choices. These details are now provided in the Methods section.

- Figure 4i. It is not clear when exactly cell activity was measured. If it was during the entire recording time, I think it would be interesting to see if the activity of O/A interneurons is different specifically during interaction with the object in 3xTg mice.

Cell activity was indeed measured throughout the entire recording session and analyzed in relation to animal behavior (immobility to walking; Fig. 4d,e), and periods specifically related to interaction with objects were extracted for analysis (Figure 4i).

- Why was the object modulation measured during a different task in which both objects were the same? The figure is misleading in that sense, as it suggests the experiment was the same as for the other panels with two different objects. It would be important to correct this if the authors want to correlate the deficits in NOR in 3xTg mice and changes in IN activity.

The study specifically investigated object-modulated neural activity during the Sampling phase. Therefore, two identical objects were placed in the arena for animal exploration. As mentioned above, due to several animals failing to explore the OFM and objects on the second day, they were excluded from the analysis, preventing the conduct of the novel-object exploration Test Trial. Both non-Tg and Tg mice showed a lack of exploration in the OFM and Tmaze, for reasons that remain unclear. Consequently, we opted to present robust data on neural activity during the initial sampling of two identical objects. However, further investigation is needed to understand how this activity relates to deficits observed in the classical NOR test.

- Figure. 5c-f. I would strongly suggest performing the same quantification and displaying similar figures for the fiber photometry experiments in interneurons and principal cells. It would help to interpret the data.

We have taken the reviewer's suggestion into account and standardized the data analysis and presentation. Figures 4d, e and 5c, d now depict the walk-induced activity in INs and PCs, respectively. Figures 4h and 5f compare activity between the stem and D-zone in the T-maze. Additionally, Figures 4j and 5h illustrate the object modulation of INs and PCs, respectively.

- Although velocity and mobility were quantified, it would be important to show also that they are not different during those times when activity was dissimilar, as in the decision zone.

We have analyzed these data and found no significant differences between the two genotypes in terms of velocity and mobility during these periods. This analysis is now presented in Supplementary Figure 5e, f and detailed in the Results section.

- Figure 5g-h. Similarly, I would suggest using the same metrics in order to correlate the results from interneuron and principal cell activity photometry.

We have updated this figure to align with the presentation of interneurons (Figure 4j) and included RMS analysis to emphasize lower variance in object modulation of PCs as an indicator of increased network inhibition.

- Was object modulation variance also different for INs depending on the mouse phenotype?

We conducted this additional analysis but did not find any significant difference.

- Figure S4: would it be possible to identify the postsynaptic partners?

As mentioned above, for those cells with preserved axonal structures, we identified both OLM and bistratified cells. We have now included this information in the Results section to clarify the types of interneurons identified.